# Depression and Anxiety in Association with Polypharmacy in Patients with Multiple Sclerosis

**DOI:** 10.3390/jcm12165379

**Published:** 2023-08-18

**Authors:** Julia Baldt, Niklas Frahm, Michael Hecker, Barbara Streckenbach, Silvan Elias Langhorst, Pegah Mashhadiakbar, Katja Burian, Janina Meißner, Felicita Heidler, Jörg Richter, Uwe Klaus Zettl

**Affiliations:** 1Section of Neuroimmunology, Department of Neurology, Rostock University Medical Centre, 18147 Rostock, Germany; niklas-frahm@gmx.de (N.F.); michael.hecker@rocketmail.com (M.H.); babswehr@web.de (B.S.); langhorst.silvan@yahoo.com (S.E.L.); pegah.mashhadiakbar@uni-rostock.de (P.M.); katja.burian@gmx.de (K.B.); janina.meissner@freenet.de (J.M.); uwe.zettl@med.uni-rostock.de (U.K.Z.); 2Ecumenic Hainich Hospital GmbH, 99974 Mühlhausen, Germany; felicita.heidler@web.de (F.H.); j.richter@oehk.de (J.R.); 3Faculty of Health Sciences, University of Hull, Hull HU6 7RX, UK; 4The Palatine Centre, Durham Law School, Durham University, Durham DH1 3LE, UK

**Keywords:** multiple sclerosis, depression, anxiety, polypharmacy, comorbidity, therapy switches

## Abstract

Polypharmacy (intake of ≥5 drugs) is an important issue for patients with chronic diseases such as multiple sclerosis (MS). We aimed to assess the prevalence of polypharmacy with regard to the severity of anxiety/depression and to comorbidities. Therefore, 374 MS patients from two German neurological sites were examined for drug burden, comorbidities, disability level and psychopathological measures capturing depression and anxiety using the Hospital Anxiety and Depression Scale (HADS-A and HADS-D). We found that patients with a higher HADS-D score take more medication (r = 0.217, *p* < 0.001). Furthermore, patients with higher depression severity were more likely to show polypharmacy (*p* < 0.001). These differences were not significant for anxiety. (*p* = 0.413). Regarding the frequency of ≥1 comorbidities, there were no significant differences between patients with different HADS-A (*p* = 0.375) or HADS-D (*p* = 0.860) severity levels, whereas the concrete number of comorbidities showed a significant positive linear correlation with HADS-A (r = 0.10, *p* = 0.045) and HADS-D scores (r = 0.19, *p* < 0.001). In conclusion, symptoms of depression pose a relevant issue for MS patients and are correlated with polypharmacy and comorbidities. Anxiety is not correlated with polypharmacy but with the frequency of several comorbidity groups in MS patients.

## 1. Introduction

Multiple sclerosis (MS) is an immune-mediated disease that affects the central nervous system [1]. Worldwide, more than 2.8 million people suffer from MS, representing the most common non-traumatic neuroimmunological disease in young adults [2,3]. MS symptoms can vary from acute unilateral optic neuritis, diplopia, nystagmus, trigeminal neuralgia, sensory loss, motor disturbances, cerebellar ataxia, urinary urge incontinence and constipation to erectile dysfunction [4,5,6,7].

Several disease-modifying drugs (DMDs) have been developed and approved for the efficient therapy of MS in recent years [8,9]. DMDs suppress or modulate the immune system and therefore are especially effective in preventing MS relapses [10,11]. Currently, nine different classes of DMDs are available [1]. Choosing the suitable DMD for the individual patient requires the consideration of comorbidities, further medications, potential drug–drug interactions, monitoring requirements and administration routes [12,13,14]. Due to disease activity despite treatment, side effects, pregnancy or personal or other reasons, there may also be a need to switch the currently used DMD [15,16].

Symptomatic drugs for MS have no direct impact on disease progression or relapse activity. However, these drugs are used to alleviate particular symptoms, such as spasticity, pain or fatigue, and to enhance the patient’s quality of life [17,18,19]. The wide range of symptoms can lead to the use of several necessary symptomatic medications.

Polypharmacy is mostly defined as the concurrent intake of at least five drugs [20,21]. The prevalence of polypharmacy in the general population of the United States was found to have increased from 7% to 19% over the 10 years from 1990 to 2000 [22]. Only a limited number of studies of polypharmacy in patients with MS are available [23,24,25]. Former studies revealed that 15% to 59% of MS patients fulfilled the criterion for polypharmacy [26]. Polypharmacy in MS patients is correlated with fatigue, pain and depression [27,28]. The risk of potential drug–drug interactions, which may lead to treatment failure and side effects, also increases with the number of medications taken [29].

The lifetime risk for depression in MS patients was reported to be 50% by Sadovnick et al., as well as Marrie and Hanwell [30,31,32]. In a 6-month follow-up investigation by Giordano et al., 11% of newly diagnosed MS patients met the criteria for a depression diagnosis [33]. A higher level of disability (as measured by the Expanded Disability Status Scale [EDSS] score) was significantly associated with the risk of suffering from a depressive disorder [34]. However, this association could not be replicated in other studies with MS patients [35,36]. A proportion of 35.6% of MS patients was found to have an anxiety disorder, whereas the prevalence of anxiety in the general population was estimated to be 29.6% [32,37]. Using the Hospital Anxiety and Depression Scale (HADS) with a cut-off of eight or more points, the proportion of MS patients suffering from anxiety varied from 14% [38] to 30.2% [39]. Higher rates and scores for depression and anxiety among radiologically active patients were found in a study by Rossi et al. analysing 105 patients with relapsing–remitting MS (RRMS) [40]. Depression and anxiety have been positively correlated with work difficulties [41] and negatively correlated with quality of life [42,43]. These findings demonstrate that anxiety and depressive disorders represent important symptoms or comorbidities in MS patients.

There have been no multi-centre studies concerning polypharmacy and depression or anxiety in MS patients. Therefore, the aim of the study was to analyse the frequency and severity of anxiety and depression in MS patients in association with polypharmacy. Thus, we analysed the frequencies and severity of depression and anxiety in MS patients. Furthermore, we aimed to identify sociodemographic and clinical variables, as well as single drugs or medication groups taken, that were associated with anxiety or depression. In a further step, the number of comorbidities in addition to MS was analysed, and we also investigated associations between the presence of comorbidity classes and anxiety or depression in MS patients.

## 2. Materials and Methods

### 2.1. Data Collection

This study was conducted in the Department of Neurology (Section of Neuroimmunology) of Rostock’s University Medical Centre (Germany) and in the Department of Neurology of the Ecumenic Hainich Hospital Mühlhausen (Germany). The study was approved by the ethics committees of the University of Rostock (permit number A 2019-0048) and of the Physicians’ Chamber of Thuringia and conducted according to the Declaration of Helsinki.

Outpatients were patients of the MS ambulatory centre of Rostock or Mühlhausen. When these patients had their regular medical appointments, they were asked to participate in the study. If patients were hospitalized for a few days, these inpatients were also asked to participate. If they agreed to participate voluntarily, patients were assigned a participant number. To ensure pseudonymization, interview sheets and questionnaires were marked with these numbers instead of the participants’ names or other identifying information. Participants were usually interviewed after their medical appointments (outpatients) or on the clinical ward (inpatients).

Data acquisition was undertaken from outpatients and inpatients who were at least 18 years of age at the time of the interview and had a diagnosis of clinical isolated syndrome (CIS) or MS (RRMS, secondary-progredient MS [SPMS] or primary progressive MS [PPMS]), according to the revised McDonald criteria [44]. The study was conducted from June 2019 to July 2020. A total of 461 patients were asked to participate voluntarily. Patients were excluded for the following reasons: unwillingness to participate, lack of time, diagnosis other than CIS or MS, inability to provide informed consent or inability to complete the HADS questionnaire on their own.

Data on socio-demographics (e.g., age, sex), MS-related clinical data (e.g., years since diagnosis, MS course type and EDSS score indicating the disability level [45]) and data on comorbid diseases, as well as the medications taken, were collected via a structured interview, patient records and a clinical examination. For further details, the interview sheet has been included as Appendix A. The EDSS is a method for quantifying disability in MS patients and ranges from 0 (no neurological deficits) to 10 (dead due to MS) in 0.5-unit increments [45,46]. Disease duration was calculated based on the date of diagnosis. The patients were asked about their medication schedules and the use of complementary medicines and dietary supplements. They were also asked about previous DMD therapy switches and switch reasons (multiple answers were possible). Finally, the participants were asked to complete the Hospital Anxiety and Depression Scale (HADS—[47]) questionnaire following the interview. The HADS-Anxiety items (HADS-A) were completed by 374 patients, and the HADS-Depression items (HADS-D) were completed by 373 patients.

The HADS has been applied as a measure of anxiety (HADS-A) and depressive symptoms (HADS-D) [47]. It was reported to be a valid screening tool with eligible psychometric properties for the diagnosis of anxiety and depressive symptoms in MS patients because somatic symptoms, such as dizziness, headache and sleep disorders, are not included as disease indicators [48]. Applying ≥8 points for depression or anxiety as a cut-off point for distinguishing between no psychopathological disturbance and any level of disturbance, sensitivity of 90% and specificity of 87% were found [49]. In a sample of MS patients, the sensitivity of the HADS-D scale (for depression) was 69%, and its specificity was 81%, whereas the sensitivity of the HADS-A scale (for anxiety) was 82%, and its specificity was 68% [50]. In the present study, the categorization of the level of anxiety/depression was performed as proposed by Marrie et al. [50]: normal scores from 0 to 7 points, borderline scores from 8 to 10 points and abnormal scores from 11 to 21 points [51,52]. The Cronbach’s alpha coefficient for the German version of the HADS questionnaire is 0.80–0.81 for both subscales. The retest stability for an interval of less than 2 weeks is greater than 0.8, and like the intended sensitivity of change, it decreases to about 0.7 for longer intervals [53].

Polypharmacy was defined as the concurrent intake of five or more medications, including over-the-counter (OTC) substances [21,26,54]. Drugs were classified according to frequency of use (regular or on-demand), prescription status (prescription or OTC drug) and treatment goal (DMD, treatment of MS symptoms or treatment of comorbidities/other conditions). Medications were classified following the Anatomical Therapeutic Chemical (ATC) classification system [55]. The number and classes of comorbid diseases were included in the data analysis. If patients reported having a comorbidity or if a current or chronic comorbidity was listed in the clinical records, we counted it as a comorbidity. If there were discrepancies, we checked with the treating neurologist. In addition, we basically followed Marrie et al. [56]. Comorbid diseases were subdivided into groups according to the organ that they normally affect; e.g., any type of thyroid disease was classified as an endocrinological disease. Only inflammatory diseases affecting more than one organ were classified as chronic inflammatory diseases, e.g., rheumatism. We defined the following groups: cardiovascular, chronic inflammatory, dermatological, endocrinological, gastrointestinal, haematological, metabolic, neurological, ophthalmologic, orthopaedic, otolaryngologic, psychiatric, pulmonary, urologic or gynaecologic diseases, pain and other diseases.

### 2.2. Statistics

Statistical analysis was performed using SPSS version 27. All patient data were pseudonymized. RRMS patients (*n* = 240) were combined with the small group of CIS patients (*n* = 19) into one group because of the frequent transition from CIS to RRMS. Interval-scaled data are presented as means and standard deviations. Student’s independent samples *t*-test, the chi-square test, the Kruskal–Wallis test and one-way analyses of variance (ANOVAs) were used to test for differences between the HADS groups. Pearson’s and Spearman’s correlation coefficients were calculated to analyse the relationships between the studied variables.

The criterion for statistical significance was *p* < 0.05.

## 3. Results

### 3.1. Study Population

Of the 461 patients who were asked to participate voluntarily, 57 patients were excluded due to unwillingness, lack of time, revision of diagnosis or inability to provide informed consent or complete the HADS questionnaire on their own. (Figure 1). In total, 374 patients (250 women and 124 men) who completed the HADS questionnaire were included in the analysis. The mean age (±standard deviation) of these patients was 48.1 ± 12.8 years old, with a mean disease duration of 12.5 ± 9.3 years. The median EDSS score as a measure of the patients’ degree of disability was 3.0. A CIS or a RRMS course type was diagnosed in 259 patients (69.3%), SPMS in 86 patients (23.0%) and PPMS in 29 patients (7.8%). The most commonly occurring comorbidities among all MS patients were hypertension (24.9%), depression (16.0%), thyroid disease (15.2%), symptoms of nutrient deficiency (11.8%), gastrointestinal symptoms (9.6%), dyslipidaemia (7.2%), bladder symptoms (6.4%), osteoporosis (5.6%), migraine (4.5%), allergy (4.5%) and bronchial asthma (4.5%).

### 3.2. Sociodemographic and Clinical Variables Related to Depression and Anxiety

A normal HADS-A score (0–7 points) was obtained for 207 patients (55.3%), a borderline score (8–10 points) for 103 patients (27.5%) and an abnormal score (11–21 points) for 64 patients (17.1%), while 264 (70.8%) patients had a normal HADS-D score (0–7 points), 71 (19.0%) a borderline score (8–10 points) and 38 (10.2%) an abnormal score (11–21 points). The relationship between HADS-A and HADS-D subgroups is shown in Figure 2. Nearly half of the patients (*n* = 183) had normal HADS-A and HADS-D scores. A minority of 5.6% (*n* = 21) had abnormal HADS-A and HADS-D scores. A normal HADS-D score but a borderline or abnormal HADS-A score occurred in 21.7% (*n* = 81) of the patients analysed.

There were no significant differences between patients with different HADS-A or HADS-D severity levels with regard to sex, age or MS course (Table 1). However, the patients’ degrees of disability were found to be significantly different depending on the severity of depressive symptoms indicated by the HADS-D, with borderline-depressed MS patients having a substantially higher median EDSS score than those in the normal group (*p* = 0.002). The time since diagnosis was substantially correlated with the anxiety level (r = −0.13, *p* = 0.012) but not with the severity of depression (r = −0.04, *p* = 0.475). Patients suffering from MS for ≤2 years (MS2y, *n* = 51) showed a significantly higher anxiety score (8.25 ± 3.77) than those who suffered for ≥10 years (MS10y, *n* = 202, 6.71 ± 3.68, T(251) = 2.66, *p* = 0.008). There was no significant difference in depression severity (MS2y vs. MS10y: 6.33 ± 4.08 vs. 5.46 ± 3.72, T(251) = 1.48, *p* = 0.141.

### 3.3. Polypharmacy and Medication Related to Severity of Depression and Anxiety

In total, 185 MS patients (49.5%) were found with polypharmacy (use of ≥5 drugs). While in the normal HADS-D group, 115 patients (43.6%) took ≥5 medications, there were 45 patients (63.4%) in the borderline HADS-D group and 25 patients (65.8%) in the abnormal HADS-D group taking that number (*p* = 0.001) (Figure 3). There was a small, but significant, correlation between the HADS-D score and the number of medications taken (r = 0.217, *p* < 0.001). When considering anxiety (HADS-A), no such difference was seen (normal vs. borderline vs. abnormal: 46.4% vs. 53.4% vs. 53.1%; r = −0.77, *p* = 0.136). Among patients with an abnormal HADS-A score and an abnormal HADS-D score (*n* = 21), 61.9% (*n* = 13) had polypharmacy.

MS patients with normal, borderline or abnormal HADS-D scores differed significantly in the mean number of drugs taken in total (*p* < 0.001) and specifically in the use of prescription drugs (*p* < 0.001), symptomatic MS drugs (*p* = 0.004) and drugs to treat comorbidities, as well as other conditions (*p* = 0.001) (Table 2). Patients in the HADS-D borderline group were on average taking significantly more different substances (6.1 drugs per patient) than those in the normal group (4.5 drugs per patient) and those in the abnormal group (5.5 drugs per patient). The same outcome was found for the mean number of drugs prescribed (borderline vs. normal vs. abnormal HADS-D: 4.9 vs. 3.6 vs. 4.3 drugs per patient). No significant differences were observed between the HADS-A patient groups in this respect (*p* ≥ 0.199).

Considering the different medication groups, vitamins (ATC classification system: A11) were used by half of the MS patients (50.0%), followed by immunosuppressants (L04—37.4%), analgesic medications (N02—27.3%), medications for acid-related disorders (A02—23.8%), immunostimulants (L03—23.3%), psychoanaleptics (N06—19.3%), corticosteroids (H02—19.0%), agents acting on the renin–angiotensin system (C09—17.6%), anti-inflammatory and antirheumatic drugs (M01—17.9%) and urological drugs (G04—16.0%). The higher the HADS-D or HADS-A score, the more often the MS patients were treated with antiparkinson drugs (N04, *p* = 0.014), psycholeptics (N05—antidementia agents, antidepressants, nootropics and stimulants, *p* = 0.001) and psychoanaleptics (N06—antipsychotics, anxiolytics, hypnotics and sedatives, *p* = 0.005) (Table 3). In addition, the patients with borderline or abnormal HADS-D scores were more likely to be taking antiepileptic drugs (N03, *p* = 0.018). Furthermore, more patients in the borderline HADS-D group were treated with drugs for functional gastrointestinal disorders (A03—2.8%, *p* = 0.014), drugs used in diabetes (A10—11.3%, *p* = 0.001), diuretics (C03—11.3%, *p* = 0.011) and lipid-modifying agents (C10—19.7%, *p* = 0.009) than those with a normal or with an abnormal HADS-D score (Table 3, Appendix A).

Excluding DMDs, the 10 most commonly used medications taken by the MS patients analysed were cholecalciferol (47.9%), pantoprazole (21.4%), ibuprofen (14.2%), cyanocobalamin (13.4%), enoxaparin (12.8%), levothyroxine (12.0%), magnesium (10.4%), baclofen (10.2%), ramipril (9.4%) and fampridine (8.0%). There were no significant differences related to the proportion of patients taking these medications depending on the HADS scores, except for pantoprazole, which was taken by significantly more patients in the borderline (32.4%) HADS-D group compared with the normal (18.6%) and abnormal HADS-D groups (21.1%) (χ^2^(2) = 6.360; *p* = 0.042).

### 3.4. Frequency of Comorbid Diseases in Relation to Depression and Anxiety

Two hundred eighty-one of the 374 patients (75.1%) had at least one comorbidity. Patients with different HADS-A (*p* = 0.375) or HADS-D (*p* = 0.860) scores did not differ significantly regarding the frequency of at least one comorbidity (Figure 3), whereas the concrete number of comorbidities showed a significant positive linear correlation with the HADS-A score (r = 0.10, *p* = 0.045) and the HADS-D score (r = 0.19, *p* < 0.001). The majority of patients with at least one comorbidity (*n* = 222, 79.0%) did not have an abnormal HADS-A or an abnormal HADS-D score. Of the 80 patients with either an abnormal HADS-A or an abnormal HADS-D score, 73.8% (*n* = 59) had at least one comorbidity (Figure 4).

The frequency of several comorbidity groups differed between patients with different levels of anxiety/depression. Neurological comorbidities were more often reported by patients with a borderline (13.6%) or abnormal (23.4%) anxiety score compared to those with normal (8.7%) scores (χ^2^(2) = 9.81, *p* = 0.007). The frequency of psychiatric comorbidities was higher in the borderline (HADS-A: 28.2%, HADS-D: 28.2%) and abnormal (HADS-A: 34.4%, HADS-D: 42.1%) groups than in the normal (HADS-A: 10.6%, HADS-D:14.0%) HADS-A and HADS-D groups (HADS-A: χ^2^(2) = 24.30, *p* < 0.001; HADS-D: χ^2^(2) = 20.77, *p* < 0.001). Metabolic comorbidities were substantially more prevalent in MS patients with borderline (35.2%) HADS-D scores compared to those in the normal (21.2%) or the abnormal (10.5%) HADS-D groups (χ^2^(2) = 9.85, *p* = 0.007). The severity of anxiety or depression was not significantly associated with any of the other comorbidity groups (cardiovascular, chronic inflammatory, dermatological, endocrinological, gastrointestinal, haematological, metabolic, neurological, ophthalmologic, orthopaedic, otolaryngologic, psychiatric, pulmonary, urologic or gynaecologic diseases, pain and other diseases; *p* ≥ 0.051). The frequencies of the most common comorbidities in our study population in association with depression and anxiety are shown in Appendix A. Depression was more frequently diagnosed in patients with high severity of anxiety, indicated by an abnormal HADS-A score (χ^2^(2) = 20.17, *p* < 0.001; normal 8.7%, borderline 22.3%, abnormal 29.7%). Exactly 50% of the patients with diagnosed depression had a normal HADS-D score, whereas 57.9% of the patients with abnormal HADS-D scores had no diagnosed depression (χ^2^(2) = 24.11, *p* < 0.001; normal 11.4%, borderline 19.7%, abnormal 42.1%). Furthermore, diagnosed migraine (*n* = 17) was significantly more common in MS patients with higher anxiety scores on the HADS-A (χ^2^(2) = 14.30, *p* = 0.001; normal 1.4%, borderline 5.8%, abnormal 12.5%). Patients suffering from an allergy (*n* = 17) were exclusively classified into the normal HADS-D group (χ^2^(2) = 7.35, *p* = 0.025; normal 6.4%, borderline 0.0%, abnormal 0.0%).

### 3.5. Associations between Switches of Disease-Modifying Therapy and Psychopathological Measures

At least one DMD therapy switch in the past was reported by 248 patients (66.3%). Fifty-five of the 80 patients with an abnormal HADS-A or HADS-D score (68.8%) reported at least one DMD switch in the past (Figure 3). However, there was no significant association between the patients’ HADS-A or HADS-D scores and the number of previous DMD switches (HADS-A: r = −0.02, *p* = 0.673, HADS-D: r = 0.08, *p* = 0.126). Regarding HADS-A, 35.7% of the normal group, 30.1% of the borderline group and 32.8% of the abnormal group never changed DMDs (Figure 3). Most of the patients in all of the HADS-A groups never changed or changed exactly once the DMD therapy (normal score: 64.7%, borderline score: 61.2%, abnormal score: 70.3%, χ^2^(10) = 5.473, *p* = 0.857). Similar findings were observed for the HADS-D groups with 65.5% of the patients in the normal, 62.0% in the borderline and 65.8% in the abnormal HADS-D groups never or only once having changed DMD therapy (χ^2^(10) = 9.921, *p* = 0.447).

It was analysed whether there were differences in the reasons for switching the DMD therapy (side effects, illness activity, pregnancy, personal reasons or other reasons) between the HADS-A and HADS-D groups.

Disease activity despite DMD treatment was the most frequently mentioned reason in the normal and in borderline HADS-A groups (mentioned by 52.6% of the patients of the normal group and 61.1% of the patients of the borderline HADS-A group). In the abnormal group, side effects and disease activity despite DMD treatment were the most commonly mentioned answers (both mentioned by 34.9% of the patients in the abnormal HADS-A group). When comparing patients stratified by HADS-A groups, there were no significant differences except for switching due to illness activity (χ^2^(2) = 7.463, *p* = 0.024). Regarding the HADS-D groups, in the normal group, the most frequently mentioned reason was disease activity (mentioned by 53.3% of the patients of the normal HADS-D group), followed by side effects, which were reported by 48.5% of patients in this HADS-D group. In the borderline and abnormal groups, side effects were the most frequently mentioned reason, followed by disease activity (borderline 52.9% vs. 51.0%, abnormal 55.6% vs. 48.1%). In all groups, pregnancy or a desire to become pregnant was a minor reason for switching (overall mentioned by 6.5% of all patients with at least one DMD switch).

## 4. Discussion

Depression and anxiety are important issues for many MS patients, as either a comorbidity or a symptom of MS, and they need to be treated adequately [57,58,59]. Because of necessary DMD treatment, symptomatic MS therapy and the need to treat comorbidities, a wide range of medications are often used in combination in MS patients. However, there have been only a few studies of polypharmacy in MS patients [24,25,26,27]. Our study is the first multi-centre study with more than 370 MS patients to investigate the associations of depression and anxiety with polypharmacy, DMD therapy switches and comorbidities in this population.

Around 30% of the MS patients analysed showed a HADS-D score of ≥8, implying that these patients had a borderline (8–10 points) or an abnormal (11–21 points) depression score. Moreover, almost every second patient suffered some extent of anxiety. In a study with 323 MS patients from Saudi Arabia, 70.0% showed a normal, 16.4% a borderline and 13.6% an abnormal HADS-D score, comparable to our findings (70.8% with normal, 19.0% with a borderline and 10.2% with an abnormal HADS-D score) [4]. Considering the anxiety level among MS patients, 57.0% of the Saudi Arabian patients had a normal HADS-A score, also similar to our findings (55.3%). However, a greater number of Saudi-Arabian patients were found to have abnormal HADS-A scores (25.7%) compared to our study population (17.1%) [4]. In another study by Giordano et al., who examined 197 MS patients at the time of diagnosis, 40% had a HADS-A score of eight or more [33]. Only 11% of the patients in the study by Giordano et al. had ≥8 points on the HADS-D scale, while 29.2% of our patients met this criterion. One reason that might explain the different distributions of HADS-A scores in the Saudi-Arabian study, as well as in the study by Giordano et al., compared to our study could be the differences in the disease durations of the study populations. In the present study, a weak, but significant, negative correlation with the anxiety score was found for the time since diagnosis. However, anxiety was not associated with the disability level as measured by the EDSS. This outcome implies that anxiety is a matter of uncertainty about the future, especially with regard to the occurrence of future MS relapses, the kind of disability and individual disease progression [60]. Giordano et al. included only patients within the first six months after MS diagnosis, whereas in the present study, the disease duration ranged up to 52 years after diagnosis [33]. The study of Saudi Arabian patients by AlSaeed et al. included exclusively patients with low disability scores and relatively short disease durations (mean number of years after the diagnosis of MS: 5.5) [4]. In another study regarding anxiety in 427 people (243 with MS, 184 without MS), there was a higher level of anxiety at the time of MS relapse compared to that in MS patients who experienced no relapses in the previous six months and that in the general population [61].

Anxiety was not found to be significantly associated with polypharmacy in our analysis. Even though more MS patients showed symptoms of anxiety than of depression, there was no significant association with the number of medications taken. However, a weak, but significant, association was found between the number of comorbidities and the severity of anxiety. In a study by Janssens et al. of 101 newly diagnosed MS patients and their partners, there was a higher level of anxiety measured by the HADS-A in patients with more functional limitations [52]. In a study with more than 7000 MS patients from the United Kingdom, female patients with RRMS were more likely to report higher levels of anxiety than female patients with PPMS or SPMS, whereas no significant association between anxiety and MS course type was found for men [62]. In contrast, in a meta-analysis of four different studies by Peres et al., there was no significant difference between the level of anxiety or depression and the course of MS [63]. Our data revealed no difference in the frequency of DMD switching depending on the level of anxiety in patients with MS. There was also no reason for prior therapy switching, which was reported significantly more often by patients with abnormal HADS-A scores. Likewise, Zanghi et al. found no difference in the reasons for the first DMD discontinuation in relation to polypharmacy [16].

Depression was diagnosed as a comorbidity in 16.0% of the patients in our study, whereas 29.2% had relevant symptoms of depression according to the HADS-D score. Only 42.1% of the MS patients with abnormal HADS-D scores were diagnosed with depression. Whether this outcome occurred because depressive symptoms were overlooked in clinical treatment or because depressive symptoms were treated with medications like other MS symptoms and not as a comorbidity cannot be assessed. Furthermore, in a study by McGuin and Hutchinson of 211 MS patients, 23.3% of the patients without diagnosed depression had relevant symptoms of depression at the time of the study [64], and in a study by Marrie et al., only one-third of the MS patients with HADS-D ≥ 8 points had diagnosed depression [65]. This finding shows that overlooking of depressive symptoms is a problem not only in our study population. The time since diagnosis did not correlate with the severity of depression. Moreover, one-fifth (19.3%) of our patients took antidepressants (ATC code N06), similar to the rate found in the study by Ploughman et al. [66]. In their study, 44.5% of 742 MS patients had either a diagnosis of depression or relevant symptoms of depression, but only 14.2% of these patients were taking antidepressants. In a study by Mohr et al., two out of three MS patients with major depression were not taking antidepressant medication [67]. This outcome is similar to a study by Cetin et al. of 542 MS patients, in which approximately 60% of the patients with relevant depressive symptoms were not taking antidepressants [68]. This finding demonstrates that depression is often overlooked and undertreated in clinical practice. This point was underscored by a study by Feinstein, who examined 140 MS patients with or without suicidal intent [69]. In that study, two out of three patients with major depression and suicidal intent were not taking antidepressants [69,70,71]. However, this outcome is in contrast to the results of a small study by Thelen et al. of 84 MS patients with polypharmacy, of whom 57% were taking antidepressants [28]. Unfortunately, the study by Thelen et al. did not report how many patients were diagnosed with a depressive disorder. It should be noted that most studies do not ask about non-medical treatments for depression, such as psychotherapy.

In our study, the total number of medications taken correlated with the degree of severity of depression according to the HADS-D (r = 0.217, *p* < 001). Polypharmacy was much more common in patients with abnormal HADS-D scores than in those with normal HADS-D scores. Patients with borderline or abnormal HADS-D scores took more medications to treat comorbidities and also more symptomatic medications, e.g., against pain or spasticity, than patients with normal HADS-D scores, confirming the findings of another study by Thelen et al. [27]. Appropriate treatment of depression can have a positive impact on the management of MS symptoms and consequently improve the patient’s quality of life. Therefore, on the one hand, active screening for depressive symptoms and appropriate therapy with avoidance of inadequate medication could represent one way to improve MS therapy management [23]. An example could be the use of antidepressants with pain-augmenting effects in depressive patients with chronic pain. On the other hand, medication with minor effects on MS symptoms or comorbidities should be consequently discontinued or substituted to avoid inappropriate polypharmacy [23]. Non-medication therapy for depression, such as psychotherapy, should be recommended. Physical exercise can improve symptoms of MS [72,73], as well as mitigate the levels of anxiety and stress [74], and may thereby reduce polypharmacy, although this relationship needs to be further investigated.

A strength of our study is the relatively large sample size, whereas the cross-sectional character represents a limitation (no statements about causal relationships could be provided). Furthermore, some subgroups, such as the abnormal HADS-D group, with a number of 38 patients, were small, implying limited validity. There are further limitations because we did not have information from all of the treating physicians (from other medical disciplines). Despite all the care taken, missing or misleading information about comorbidities or medications was theoretically possible. In addition, only patients who were inpatients or outpatients at one of the two MS centres were included. Therefore, it is not possible to say anything about the treatment of patients seen by neurologists in private practice, introducing selection bias. Follow-up studies gathering longitudinal data should be performed to investigate the courses of depression and anxiety among MS patients and the temporal relations with polypharmacy, comorbidities and disease duration.

In conclusion, we performed the first study exploring the relationship between polypharmacy and anxiety, as well as depression, in patients with MS. While there was a significant association of polypharmacy with the depression severity level, there was no significant association with anxiety. We found evidence that higher levels of anxiety were present in the first years after the diagnosis of MS, whereas depression was not associated with MS disease duration. However, psychopathological measures were not related with switching DMD therapies. Physicians should pay special attention to symptoms of anxiety and depression to recognize them as true health problems in patients with MS. Inappropriate polypharmacy should be avoided in the therapeutic management of MS. Further studies, particularly longitudinal studies, are necessary to investigate the influence of the consistent treatment of anxiety and depression on polypharmacy rates and patient quality of life.

## Figures and Tables

**Figure 1 jcm-12-05379-f001:**
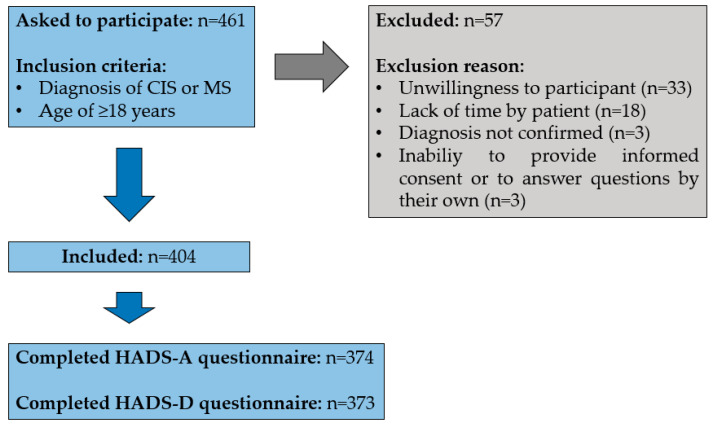
Patient selection. Inclusion and exclusion of patients are shown. CIS—Clinical Isolated Syndrome, HADS-A—subscale of anxiety of the Hospital Anxiety and Depression Scale, HADS-D—subscale of depression of the Hospital Anxiety and Depression Scale, MS—multiple Sclerosis, *n*—number of patients.

**Figure 2 jcm-12-05379-f002:**
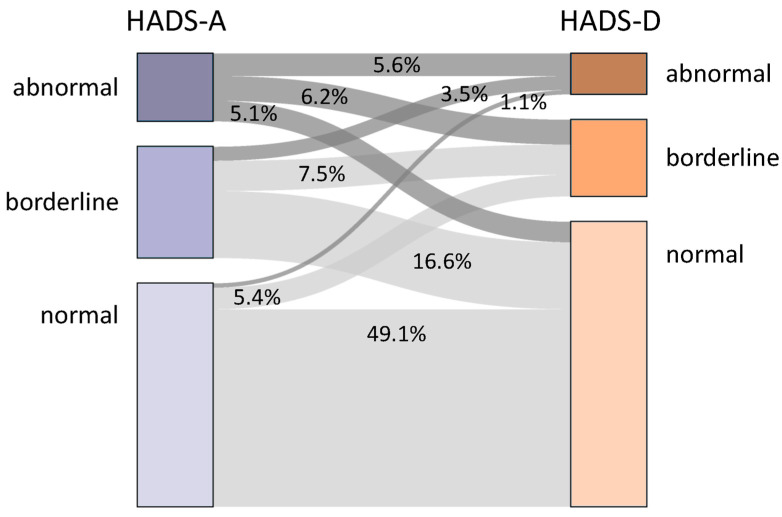
Proportion of patients in the different HADS-A and HADS-D subgroups. The distribution of the patients between the HADS categories of the 373 patients who completed both questionnaires is shown. Almost half (49.1%) of the patients with multiple sclerosis had a normal HADS-A score and a normal HADS-D score. HADS-A—subscale of anxiety of the Hospital Anxiety and Depression Scale, HADS-D—subscale of depression of the Hospital Anxiety and Depression Scale.

**Figure 3 jcm-12-05379-f003:**
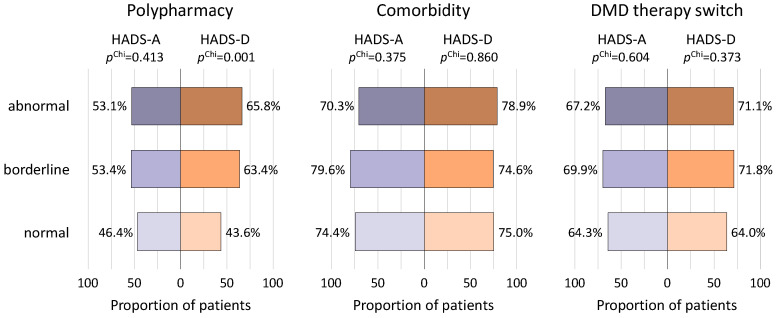
Polypharmacy, comorbidity and DMD therapy switches in association with symptoms of anxiety and depression. For each HADS-A and HADS-D category, the proportions of patients with polypharmacy, comorbidity and DMD therapy switches are visualized. Regarding the prevalence of polypharmacy (i.e., intake of at least five drugs), there was a significant difference between the different HADS-D groups (*p* = 0.001) but not between the different HADS-A groups (*p* = 0.413). There was no significant difference for having at least one comorbidity (i.e., diagnosis other than MS) or at least one DMD therapy switch (i.e., change in the use of DMDs in the past) between patients stratified by anxiety or depression severity level (*p* ≥ 0.373). Chi—chi-square test, DMD—disease-modifying drug, HADS-A—subscale of anxiety of the Hospital Anxiety and Depression Scale, HADS-D—subscale of depression of the Hospital Anxiety and Depression Scale, MS—multiple sclerosis; *p*—*p*-value for comparing patients with different HADS-A or HADS-D scores.

**Figure 4 jcm-12-05379-f004:**
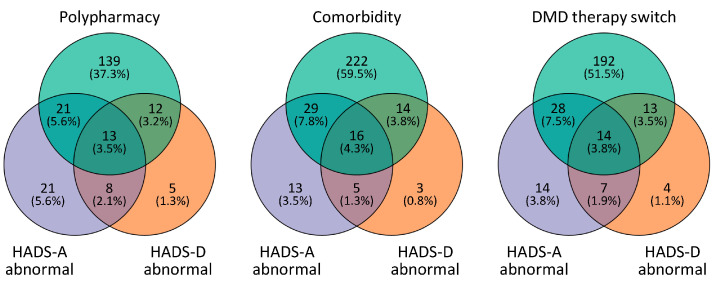
Intersection of patients with an abnormal HADS-A score or an abnormal HADS-D score and those who had polypharmacy, a comorbidity or a previous DMD therapy switch. Shown are numbers and percentages of patients of all the patients with complete data (*n*= 373) who met 1, 2 or 3 particular criteria. For instance, while 37.3% (*n* = 139) of the patients fulfilled the criterion for polypharmacy and had normal or borderline HADS-A and HADS-D scores, 3.5% (*n* = 13) fulfilled the criterion for polypharmacy and had an abnormal HADS-A score, as well as an abnormal HADS-D score (left Venn diagram). A subset of 12.1% (*n* = 45) of the patients had an abnormal HADS-A score and at least one comorbidity (central Venn diagram). One hundred ninety-two (51.5%) of the patients who switched their DMD therapy at least once previously had neither an abnormal HADS-A score nor an abnormal HADS-D score (right Venn diagram). DMD—disease-modifying drug, HADS-A—subscale of anxiety of the Hospital Anxiety and Depression Scale, HADS-D—subscale of depression of the Hospital Anxiety and Depression Scale, *n*—number of patients.

**Table 1 jcm-12-05379-t001:** Sociodemographic and clinical data of the patients with MS stratified by HADS scores.

	HADS-A				HADS-D			
**HADS score**	0–7	8–10	11–21	*p*	0–7	8–10	11–21	*p*
***n* (%)**	207 (55.3)	103 (27.5)	64 (17.1)		264 (70.8)	71 (19.0)	38 (10.2)	
**Sex, *n* (%)**				0.239 ^1^				0.090 ^1^
Female	132 (52.8)	70 (28.0)	48 (19.2)		184 (73.9)	45 (18.1)	20 (8.0)	
Male	75 (60.5)	33 (26.6)	16 (12.9)		80 (64.5)	26 (21.0)	18 (14.5)	
**Age (years), mean ± SD**	49.3 ± 12.2	46.7 ± 13.6	46.7 ± 13.4	0.161 ^2^	47.6 ± 12.6	50.6 ± 14.1	47.3 ± 11.7	0.199 ^2^
**EDSS score, median**	3.5	3.0	3.0	0.317 ^3^	3.0	4.0	3.25	**0.002** ^3^*
**Disease duration (years), mean ± SD**	13.2 ± 8.8	12.1 ± 9.4	10.9 ± 10.8	0.194 ^2^	12.7 ± 9.2	13.5 ± 10.1	9.6 ± 8.4	0.097 ^2^
**MS course, *n* (%)**				0.287 ^1^				0.128 ^1^
CIS/RRMS	134 (51.7)	76 (29.3)	49 (18.9)		189 (73.3)	40 (15.5)	29 (11.2)	
SPMS	56 (65.1)	19 (22.1)	11 (12.8)		56 (65.1)	23 (26.7)	7 (8.1)	
PPMS	17 (58.6)	8 (27.6)	4 (13.8)		19 (65.5)	8 (27.6)	2 (6.9)	

^1^—chi-square test, ^2^—one-way analysis of variance, ^3^—Kruskal–Wallis test, bold *p*—*p* < 0.05, *—*p* after FDR correction < 0.05, CIS—clinically isolated syndrome, df—degree of freedom, DMD—disease-modifying drug, EDSS—Expanded Disability Status Scale, FDR—false discovery rate, HADS-A—subscale of anxiety of the Hospital Anxiety and Depression Scale, HADS-D—subscale of depression of the Hospital Anxiety and Depression Scale, *n*—number of patients, *p*—*p*-value for comparing patients with different HADS-A or HADS-D score, PPMS—primary progressive multiple sclerosis, RRMS—relapsing–remitting multiple sclerosis, SD—standard deviation, SPMS—secondary progressive multiple sclerosis.

**Table 2 jcm-12-05379-t002:** Medication data of MS patients stratified by HADS groups.

	HADS-A				HADS-D			
HADS score	0–7	8–10	11–21	*p*	0–7	8–10	11–21	*p*
*n* (%)	207 (55.3)	103 (27.5)	64 (17.1)		264 (70.8)	71 (19.0)	38 (10.2)	
Number of drugs taken, mean ± SD	4.8 ± 2.8	5.1 ± 3.0	5.1 ± 3.3	0.475	4.5 ± 2.6	6.1 ± 3.6	5.5 ± 3.1	**<0.001** ***
Prescription drugs, mean ± SD	3.8 ± 2.6	4.0 ± 2.8	4.0 ± 3.0	0.757	3.6 ± 2.4	4.9 ± 3.4	4.3 ± 3.0	**<0.001** **
Over-the-counter drugs, mean ± SD	1.0 ± 1.2	1.1 ± 1.2	1.2 ± 1.1	0.430	1.0 ± 1.2	1.3 ± 1.2	1.2 ± 1.3	0.157
DMD, mean ± SD	0.8 ± 0.4	0.8 ± 0.4	0.7 ± 0.5	0.224	0.8 ± 0.4	0.7 ± 0.4	0.7 ± 0.5	0.221
Symptomatic MS drugs, mean ± SD	1.9 ± 1.9	2.0 ± 2.0	1.8 ± 1.9	0.774	1.7 ± 1.8	2.4 ± 1.9	2.5 ± 2.6	**0.004** **
Drugs to treat comorbidities and other conditions, mean ± SD	2.1 ± 2.0	2.4 ± 2.2	2.6 ± 2.6	0.199	2.0 ± 1.9	3.1 ± 2.8	2.1 ± 2.0	**0.001** **

One-way analysis of variance was used to calculate *p*-values. bold *p*—*p* < 0.05, **—*p* after FDR correction < 0.01, ***—*p* after FDR correction < 0.001 CIS—clinically isolated syndrome, df—degree of freedom, DMD—disease-modifying drug, EDSS—Expanded Disability Status Scale, FDR—false discovery rate, HADS-A—subscale of anxiety of the Hospital Anxiety and Depression Scale, HADS-D—subscale of depression of the Hospital Anxiety and Depression Scale, *n*—number of patients, *p*—*p*-value for comparing patients with different HADS-A or HADS-D score, PPMS—primary progressive multiple sclerosis, RRMS—relapsing–remitting multiple sclerosis, SD—standard deviation, SPMS—secondary progressive multiple sclerosis.

**Table 3 jcm-12-05379-t003:** Number of patients taking several medication classes according to ATC classification system groups stratified by the patients’ level of depression and anxiety.

	HADS-A					HADS-D				
HADS score	0–7	8–10	11–21	Total sample	*p*	0–7	8–10	11–21	Total sample	*p*
*n* (%)	207 (55.3)	103 (27.5)	64 (17.1)	374 (100.0)		264 (70.8)	71 (19.0)	38 (10.2)	373 (100.0)	
A03—Drugs for functional gastrointestinal disorders	0 (0.0)	1 (1.0)	1 (1.6)	2 (0.5)	0.253	0 (0.0)	2 (2.8)	0 (0.0)	2 (0.5)	**0.014**
A10—Drugs used in diabetes	7 (3.4)	3 (2.9)	4 (6.3)	14 (3.7)	0.499	5 (1.9)	8 (11.3)	1 (2.6)	14 (3.8)	**0.001 ***
C03—Diuretic drugs	10 (4.8)	4 (3.9)	3 (4.7)	17 (4.5)	0.930	8 (3.0)	8 (11.3)	1 (2.6)	17 (4.6)	**0.011**
C10—Lipid-modifying agents	22 (10.6)	12 (11.7)	5 (7.8)	39 (10.4)	0.725	24 (9.1)	14 (19.7)	1 (2.6)	39 (10.5)	**0.009**
N03—Antiepileptic drugs	28 (13.5)	14 (13.6)	8 (12.5)	50 (13.2)	0.975	28 (10.6)	12 (16.9)	10 (26.3)	50 (13.4)	**0.018**
N04—Antiparkinson drugs	2 (1.0)	4 (3.9)	5 (7.8)	11 (2.9)	**0.014**	4 (1.5)	4 (5.6)	3 (7.9)	11 (2.9)	**0.031**
N05—Psycholeptic drugs	3 (1.4)	3 (2.9)	7 (10.9)	13 (3.5)	**0.001 ***	5 (1.9)	3 (4.2)	5 (13.2)	13 (3.5)	**0.002**
N06—Psychoanaleptics	28 (13.5)	25 (24.3)	19 (29.7)	72 (19.5)	**0.005**	40 (15.2)	17 (23.9)	15 (39.5)	72 (19.3)	**<0.001 ***
V03—All other therapeutic products	0 (0.0)	0 (0.0)	2 (3.1)	2 (0.5)	**0.008**	0 (0.0)	2 (2.8)	0 (0.0)	2 (0.5)	**0.014**

The chi-square test was used to calculate *p*-values. Only ATC codes for which the difference between the groups reached the statistical significance level are shown. The full results are provided in Appendix A. bold *p*—*p* < 0.05, *—*p* after FDR correction < 0.05, ATC—Anatomical Therapeutic Chemical classification system, FDR—false discovery rate, HADS-A—subscale of anxiety of the Hospital Anxiety and Depression Scale, HADS-D—subscale of depression of the Hospital Anxiety and Depression Scale, *n*—number of patients, *p*—*p*-value for comparing patients with different HADS-A or HADS-D score.

## Data Availability

The datasets generated and analysed in the current study are available from the corresponding author on reasonable request.

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
