# Peer review of "Depression and Anxiety in Association with Polypharmacy in Patients with Multiple Sclerosis"

_jcm, 2023, doi:10.3390/jcm12165379_

Round 1

Reviewer 1 Report

an interesting study with important findings: polypharmacy in MS trlated to depression and anxiety. Well examined in a large number of patients (>350) from two sites, using adequate measures, apllying adequate statistics leading to convincing results and carrying out thoughtful discussions

Author Response

Response to Reviewer 1 Comments

an interesting study with important findings: polypharmacy in MS trlated to depression and anxiety. Well examined in a large number of patients (>350) from two sites, using adequate measures, apllying adequate statistics leading to convincing results and carrying out thoughtful discussions

Response: We thank the reviewer for evaluating our manuscript.

Reviewer 2 Report

I would like to thanks for the opportunity to review the manuscript entitled Depression and anxiety in association with polypharmacy in 2 patients with multiple sclerosis.

The manuscript is understandable and introduces important aspects of assessment of the frequency and severity of anxiety and depression in MS patients in association to polypharmacy.

However, the following points should be addressed:

In data collection section the authors should:

1. explain how and where the questionnaires were administered and how the anonymous was ensured;

2. specify the inclusion criteria;

3. add information about Cronbach’s alpha coefficient for German version of questionnaire which were used in the study.

Author Response

I would like to thanks for the opportunity to review the manuscript entitled Depression and anxiety in association with polypharmacy in 2 patients with multiple sclerosis.

The manuscript is understandable and introduces important aspects of assessment of the frequency and severity of anxiety and depression in MS patients in association to polypharmacy.

However, the following points should be addressed:

In data collection section the authors should:

  1. explain how and where the questionnaires were administered and how the anonymous was ensured;

Response:

Dear Reviewer, outpatients were patients of the MS ambulance of Rostock or Mühlhausen. When these patients had their regular medical appointment, they were asked to take part in the study. If patients were hospitalized of a few days, these inpatients were also asked to participate. If they agreed to participate voluntarily, patients were assigned a participant number. To ensure pseudonymization, interview sheets and questionnaires were marked with these numbers instead of the participants’ names or other identifying information. Participants were usually interviewed after their medical appointment (outpatients) or on the clinical ward (inpatients). They completed the HADS questionnaire immediately after the interview. In order to link the pseudonym numbers to the correct patients, lists were created in Mühlhausen and Rostock and remained in the clinics for security reasons. (see pages 2-3, lines 91-98)

  1. specify the inclusion criteria;

Response:

Dear Reviewer, the inclusion criteria were age ≥18years and diagnosis of CIS or MS according to McDonald criteria of 2017, regardless of type of MS, EDSS or disease duration. We specified this statement in the manuscript (page 3, lines 99-102) and added a flowchart at the beginning of the results section (see Figure 1, page 5).

  1. add information about Cronbach’s alpha coefficient for German version of questionnaire which were used in the study.

Response:

Dear Reviewer, The Cronbach’s alpha coefficient for the German version of the HADS questionnaire is 0.80-0.81 for both subscalas. The retest stability for an interval less than two weeks is above 0.8 and like the intended sensivity of change decreases to about 0.7 for longer intervals. We added this in the manuscript (see page3, lines 134-137)

Reviewer 3 Report

Introduction

  1. The introduction is clear and concise, and the problem is well stated and framed. However, (and see below) the objective of identifying associations is not adequately developed.

Methods

  1. A better description of the inclusion and exclusion criteria should be provided. How and by whom was “poor cognitive state” defined  in order to exclude the patients?

  2. The number of patients screened, excluded (and their reasons) and finally included in the analysis should be presented in the results section. A flowchart would be very informative.

  3. Why include years since diagnosis and not years since the first symptom?

  4. A better description of the structured interview should be provided. How were comorbidities classified in this interview? Has it been validated previously? How were discrepancies between the interview, clinical records and examination managed and was there a pre-specified plan for this case?

  5. The description of how many patients answered the HADS should be in the results section.

  6. Were the current DMT included for the definition of polypharmacy?

  7. How were the comorbidities classified? Further detail is warranted. For example, uveitis is both an ophthalmological and inflammatory disease, and autoimmune thyroiditis is also inflammatory but has metabolic manifestations.

  8. Were subgroup analyses performed?

  9. Regardless of the study being exploratory, the p-values and their interpretation should be corrected for multiple comparisons. Otherwise, false conclusions of correlations may be reached.

  10. How did the authors assess and correct for multicollinearity between variables and confounding?

Results

  1. As said, a flowchart of the inclusion and exclusion would be very valuable.

  2. Is the disease duration reported calculated based on the time of diagnosis or symptomatic onset?

  3. On the EDSS, it is usually non-normally distributed, so mean and SD are not the ideal descriptive statistics. It is also well known for neurologists involved in MS care, but, since the focus of the journal is not MS neurologists, a brief description with reference should be provided in the methods section (as done for the HADS).

  4. As understood from the methods section, the study only involves patients with CIS and RRMS, but ⅓ have primary and secondary progressive MS. As said, inclusion and exclusion criteria should be more clearly defined in the text. Also, as said above, a brief description of the classification used should be provided in the methods section (with its adequate reference).

  5. Regarding the comorbidities described, there is a high frequency of symptoms (of nutrient deficiency, gastrointestinal, bladder). This is problematic for several reasons; for example bladder symptoms may be secondary to MS itself and different nutrient deficiencies have a wide array of symptoms, difficult to classify as one single disease. Again, the process of classification of comorbidities should be more clearly explained.

  6. In the results, the authors state that 374 patients completed the HADS and the analysis is based on this number. However, in the methods section it is stated that 373 completed the HADS-D section. Why this difference and how was it interpreted?

  7. The Sankey diagram provided in figure 1 is very attractive and depicts the relationships between HADS D and HADS A. However, its interpretation is flawed. The percentages should be depicted based not on the total number of patients, but on the column on the left (in this case, HADS-A). The legend in figure 1 repeats what was written in the text; it should explain the figure for clarification. Further, although a small margin, the percentages presented sum over 100%.

  8. Table 1, 2, and 3 are very cluttered. I would suggest dropping some data that may be redundant (for example, the test score/degrees of freedom and p-value are expressions of the same information). “Chi” and “ANO” in superscript could be replaced with numbers, letters or symbols; I don’t know if there is a special instruction from the journal in this regard. Also, for the continuous variables the SD should be checked; is it +/- or is the number the actual SD? In table 2, the line on the number of DMDs leads to the conclusion that some patients are using more than one; is this currently or is this historical?. If currently, how many and why are using more than one DMD?

  9. Regarding the medication groups, what immunosupressants/antiinflamatory/antirheumatic drugs were the patients taking? Were these the DMDs? The same goes to corticosteroids, which are not commonly used for treating MS chronically; what were their indications?

  10. The authors state that “patients with borderline or abnormal HADS-D scores took more often antiepileptic drugs”. This may lead to conclude that the frequency of epilepsy is high, but it is actually never reported. This raises the question of the actual indications of the medications taken (for example, antiepileptics or antidepressants for headache).

  11. Regarding the DMD switches, since they occurred necessarily in the past, the time between them and the study should be included in the analysis. Is the 6.5% of the patients who switched for pregnancy related reasons based on the total sample or only on the women?

Discussion

  1. How do the authors interpret the fact that medication use is not linear in relation to the severity of depression/anxiety (as would be expected) but rather peaks in the borderline groups?

  2. The discrepancy between the frequency of reported depression and depressive symptoms is relevant and should be discussed further.

  3. There are several other limitations of this study that should be discussed. Just to mention a few, the sources of information, selection and participation bias and multiple comparisons challenge the validity of the results.

  4. Although the authors present the study as merely exploratory, the discussion is full of statements suggesting associations between depression/anxiety and polypharmacy. This is frankly misleading, since the statistical methods are not enough to support those statements. This should be carefully reviewed by the authors.

Overall:

I would suggest to adhere to the STROBE guidelines on reporting of observational studies.

Needs to be improved. 

Author Response

Introduction

  1. The introduction is clear and concise, and the problem is well stated and framed. However, (and see below) the objective of identifying associations is not adequately developed.

Response:

Dear Reviewer, we presented the objectives more detailly (page 2, lines 74-82).

Methods

  1. A better description of the inclusion and exclusion criteria should be provided. How and by whom was “poor cognitive state” defined in order to exclude the patients?

Response:

We thank the reviewer for pointing this out. Inclusion criteria were MS patients ≥18 years of age diagnosed with CIS or MS according to the McDonald criteria of 2017, regardless of MS type, EDSS or disease duration. Another requirement was that the patients were able to provide informed consent and participate in the interview as well as complete the questionnaires independently. If this was not possible due to cognitive deficits patients were excluded. We concretised this statement in the manuscript (see page3, lines 99-106). Additionally, we added a flowchart with inclusion and exclusion criteria in the result section (see page 5, Figure 1)

  1. The number of patients screened, excluded (and their reasons) and finally included in the analysis should be presented in the results section. A flowchart would be very informative.

Response:

Dear Reviewer, we added a flowchart with the inclusion and exclusion criteria and the number of patients screened at the beginning of the results section (see page 5, Figure 1)

  1. Why include years since diagnosis and not years since the first symptom?

Response:

Dear Reviewer, the date of diagnosis is documented regularly in the patients’ clinical records and is thus clinically confirmed. Many patients do not know the exact date of onset of their symptoms and this information was only sporadically documented in the patient records. Therefore, we used the month of diagnosis to calculate disease duration.

  1. A better description of the structured interview should be provided. How were comorbidities classified in this interview? Has it been validated previously? How were discrepancies between the interview, clinical records and examination managed and was there a pre-specified plan for this case?

Response:

Dear Reviewer, we added the interview sheet as Supplementary Document S1.

If the patients reported having the comorbidity or if a current or chronic comorbidity was listed in the clinical records, we counted it as a comorbidity. If there were discrepancies, we checked this together with the treating neurologist. We revised the methods section accordingly (page 3, lines 144-147).

  1. The description of how many patients answered the HADS should be in the results section.

Response:

Dear Reviewer, we present the numbers of patients in the flowchart at the beginning of the results section (see page 5, Figure 1).

  1. Were the current DMT included for the definition of polypharmacy?

Response:

Dear Reviewer, every drug taken was included to assess polypharmacy, as well as the DMT (page 3, lines 138-139).

  1. How were the comorbidities classified? Further detail is warranted. For example, uveitis is both an ophthalmological and inflammatory disease, and autoimmune thyroiditis is also inflammatory but has metabolic manifestations.

Response:

Dear Reviewer, comorbidities were classified by the affected organ, because it is more consistent. We added a statement in the methods section (see page 3, lines 147-150).

  1. Were subgroup analyses performed?

Response:

Dear Reviewer, analyses for the HADS-A and HADS-D subgroups were conducted as shown in the manuscript. Further subgroup analyses were not performed.

  1. Regardless of the study being exploratory, the p-values and their interpretation should be corrected for multiple comparisons. Otherwise, false conclusions of correlations may be reached.

Response:

Dear Reviewer, we added the False Discovery Rate (FDR) correction for multiple comparisons (see Tables 1-3, Supplementary Tables 1-3).

  1. How did the authors assess and correct for multicollinearity between variables and confounding?

Response:

Dear Reviewer, since we did not apply multiple regression, there was no need to test for multicollinearity at all.

Results

  1. As said, a flowchart of the inclusion and exclusion would be very valuable.

Response:

Dear Reviewer, we added a flowchart at the beginning of the results section (see page 5, Figure 1).

  1. Is the disease duration reported calculated based on the time of diagnosis or symptomatic onset?

Response:

Dear Reviewer, the disease duration is reported based on the date of diagnosis (see page 3, lines 114-115).

  1. On the EDSS, it is usually non-normally distributed, so mean and SD are not the ideal descriptive statistics. It is also well known for neurologists involved in MS care, but, since the focus of the journal is not MS neurologists, a brief description with reference should be provided in the methods section (as done for the HADS).

Response:

Dear Reviewer, we added a short description in the methods section (see page 3, lines 112-114).

In the results and Table 1, we changed the mean to the median value (see Table 1 and page 4, lines 173-174).

  1. As understood from the methods section, the study only involves patients with CIS and RRMS, but ⅓ have primary and secondary progressive MS. As said, inclusion and exclusion criteria should be more clearly defined in the text. Also, as said above, a brief description of the classification used should be provided in the methods section (with its adequate reference).

Response:

Dear Reviewer, we added the inclusion and exclusion criteria in a flowchart at the beginning of the results section (see page 5, Figure 1). All MS patients were included regardless of the course type. We have specified this in the methods section (see page 3, lines 99-102)

  1. Regarding the comorbidities described, there is a high frequency of symptoms (of nutrient deficiency, gastrointestinal, bladder). This is problematic for several reasons; for example bladder symptoms may be secondary to MS itself and different nutrient deficiencies have a wide array of symptoms, difficult to classify as one single disease. Again, the process of classification of comorbidities should be more clearly explained.

Response:

Dear Reviewer, if the patients said they had the comorbidity or if a current or chronic comorbidity was listed in the medical records, we counted it as a comorbidity. In addition, if there were discrepancies, we checked this with the treating neurologist. The methods section was expanded accordingly (page 3, lines 144-147). In the case of low vitamin levels, the comorbidity was classified as a nutrient deficiency. Furthermore, we basically followed Marrie et al. (Neurology, 2016, PMID 26865523) for the definition of comorbidities.

  1. In the results, the authors state that 374 patients completed the HADS and the analysis is based on this number. However, in the methods section it is stated that 373 completed the HADS-D section. Why this difference and how was it interpreted?

Response:

Dear Reviewer, the HADS questionnaire consists of 14 questions (7 regarding symptoms of depression and 7 regarding symptoms of anxiety). These questions are alternated. One patient did not answer the last (depression) question of the questionnaire. Without 1 out of 7 questions, the score for the HADS-D could not be determinable. The missing answer does not affect the HADS-A score. Therefore, we did not exclude the patient at all, but we did not use the information of the HADS-D. As a result, 374 patients were included, with 374 HADS-A scores but only 373 HADS-D scores.

  1. The Sankey diagram provided in figure 1 is very attractive and depicts the relationships between HADS D and HADS A. However, its interpretation is flawed. The percentages should be depicted based not on the total number of patients, but on the column on the left (in this case, HADS-A). The legend in figure 1 repeats what was written in the text; it should explain the figure for clarification. Further, although a small margin, the percentages presented sum over 100%.

Response:

Dear Reviewer, only 373 patients completed both the HADS-A and the HADS-D questionnaires. Therefore, the percentages are based on 373 and not on 374 patients. This issue and also rounding differences explain the sum of more than 100%. We added an information in the legend of Figure 2. (see Figure 2, page 6)

  1. Table 1, 2, and 3 are very cluttered. I would suggest dropping some data that may be redundant (for example, the test score/degrees of freedom and p-value are expressions of the same information). “Chi” and “ANO” in superscript could be replaced with numbers, letters or symbols; I don’t know if there is a special instruction from the journal in this regard. Also, for the continuous variables the SD should be checked; is it +/- or is the number the actual SD? In table 2, the line on the number of DMDs leads to the conclusion that some patients are using more than one; is this currently or is this historical?. If currently, how many and why are using more than one DMD?

Response:

Dear Reviewer, we dropped the information about the test-score and the degree of freedom and replaced “Chi” and “ANO” by numbers (see Tables 1-3, Supplementary tables 1-3). Standard deviation is a measure of how the data are spread around the mean. A low standard deviation indicates that data are clustered around the mean; high standard deviation indicates data are more spread out. Therefore, the standard deviation is reported with ± (https://www.nlm.nih.gov/nichsr/stats_tutorial/section2/mod8_sd.html). Table 2 revealed how many DMDs the patients were currently taking (range: 0 to 1).

  1. Regarding the medication groups, what immunosupressants/antiinflamatory/antirheumatic drugs were the patients taking? Were these the DMDs? The same goes to corticosteroids, which are not commonly used for treating MS chronically; what were their indications?

Response:

We thank the reviewer for this point. The medication groups were classified according to the ATC code. If a corticosteroid was being used at the time of the interview, the drug was classified as used, prescribed drug. Systemic corticosteroids were classified as ATC group H02. Drugs of the group H are systemic hormonal preparations, excl. sex hormons and insulins. Interferon beta and glatiramer acetate were classified as L03-Immunostimulants in the group of L- antineoplastic and immunomodulating agents. Dimethyl fumarate, teriflunomide, cladribine, ocrelizumab, alemtuzumab, natalizumab, azathioprine and methotrexate were classified as L04 - Immunosuppressants. Mitoxantrone was classified as L01 antineoplastic agents. Because the ATC distinguishes formally by agent and only partially by indication, it is not possible to directly infer the indication from the ATC code.

However, we classified the drugs taken according to the treatment goal (DMD to treat MS, symptomatic treatment of MS, therapy of comorbidities or other conditions). This information was collected through the structured interview or patient records. We revised the methods section accordingly (see page 3, lines 139-142).

To present the distribution of indications for the use of corticosteroid to treat comorbidities: In our population, one corticosteroid user had rheumatism, two patients had vasculitis, one patient had lupus erythematosus, four patients had bronchial asthma and one patient had Crohn`s disease.

  1. The authors state that “patients with borderline or abnormal HADS-D scores took more often antiepileptic drugs”. This may lead to conclude that the frequency of epilepsy is high, but it is actually never reported. This raises the question of the actual indications of the medications taken (for example, antiepileptics or antidepressants for headache).

Response:

Dear Reviewer, the medication was classified by the ATC code. This classification does not only follow the indication. As a consequence, it is not possible to directly link the use of antiepileptic drugs to the incidence of epilepsy. For example, some antiepileptics may be used to treat migraine.

  1. Regarding the DMD switches, since they occurred necessarily in the past, the time between them and the study should be included in the analysis. Is the 6.5% of the patients who switched for pregnancy related reasons based on the total sample or only on the women?

Response:

Dear reviewer, the time from DMD switch to the study was have not been acquired, so we are not able to provide this information. Reasons for DMD switch were collected from all patients, regardless of sex. Therefore, the percentages of reasons given refer to the total number of patients with at least one DMD switch.

Discussion

  1. How do the authors interpret the fact that medication use is not linear in relation to the severity of depression/anxiety (as would be expected) but rather peaks in the borderline groups?

Response:

Dear Reviewer, the abnormal HADS-D group take more medications than the normal HADS-D group, but less than the borderline HADS-D group. Considering the correlation between the HADS-D score and the number of drugs taken, there is a small, but significant correlation (r=0,217, p<0.001).The abnormal HADS-D group is very small (only 38 patients). From a statistical point of view, this could be one reason, why the total drug intake was somewhat lower. A statement has already been included in the limitations section (page 14, line 495-496).

  1. The discrepancy between the frequency of reported depression and depressive symptoms is relevant and should be discussed further.

Response:

Dear Reviewer, we added further studys with the same discrepancy between reported depressive symptoms and the diagnosis of depression and expanded this point of discussion (see page 13-14, lines 451-463).

  1. There are several other limitations of this study that should be discussed. Just to mention a few, the sources of information, selection and participation bias and multiple comparisons challenge the validity of the results.

Response:

Dear Reviewer, we revised our limitations (see page 14, lines 493-505).

  1. Although the authors present the study as merely exploratory, the discussion is full of statements suggesting associations between depression/anxiety and polypharmacy. This is frankly misleading, since the statistical methods are not enough to support those statements. This should be carefully reviewed by the authors.

Response:

Dear Reviewer, our analysis is a large study with 374 patients. We found associations between depression/anxiety and polypharmacy, but the lack of longitudinal data does not allow us to make causal statements. Because of the large study population and the study design, statements about associations are possible. We reviewed the wording thoughout the discussion to ensure that there was no wording that might imply a causal relationship.

Overall:

I would suggest to adhere to the STROBE guidelines on reporting of observational studies.

Response:

Dear Reviewer, we followed the strobe guidelines.

As suggested, we have once again revised the entire manuscript with regard to the English language.